# Cerebrospinal Fluid Dynamics and Partial Pressure of Carbon Dioxide as Prognostic Indicators in Hypoxic–Ischemic Brain Injury Following Cardiac Arrest

**DOI:** 10.3390/brainsci14030297

**Published:** 2024-03-21

**Authors:** So Young Jeon, Hong Joon Ahn, Changshin Kang, Yeonho You, Jung Soo Park, Jin Hong Min, Wonjoon Jeong, Yong Nam In

**Affiliations:** 1Department of Emergency Medicine, Chungnam National University Hospital, 282 Munhwa-ro, Jung-gu, Daejeon 35015, Republic of Korea; chloe9899@naver.com (S.Y.J.); rosc@cnu.ac.kr (C.K.); yyo1003@naver.com (Y.Y.); cpcr@cnu.ac.kr (J.S.P.); gardenjun@hanmail.net (W.J.); 2Department of Emergency Medicine, College of Medicine, Chungnam National University School of Medicine, Daejeon 35015, Republic of Korea; laphir2006@naver.com (J.H.M.); ynsoft@naver.com (Y.N.I.); 3Department of Emergency Medicine, Chungnam National University Sejong Hospital, 20 Bodeum 7-ro, Sejong 30099, Republic of Korea

**Keywords:** cardiac arrest, cerebrospinal fluid, partial pressure of carbon dioxide, prognostication, neurological recovery, cerebrospinal fluid dynamics, hypoxic–ischemic brain injury, cerebral hypoxia, out-of-hospital cardiac arrest

## Abstract

Changes in cerebrospinal fluid (CSF) dynamics can have adverse effects on neuronal function. We hypothesized that patients with hypoxic–ischemic brain injury (HIBI) showing poor neurological outcomes after cardiac arrest (CA) would exhibit changes in CSF dynamics, leading to abnormalities in gas diffusion within the CSF. Therefore, we investigated the prognostic value of the CSF partial pressure of carbon dioxide (PcsfCO_2_) in CA survivors who underwent targeted temperature management (TTM). We retrospectively analyzed the 6-month neurological outcomes, CSF, and arterial blood gas parameters of 67 CA survivors. Patients were divided into good and poor neurological outcome groups, and the predictive value of PcsfCO_2_ for poor neurological outcomes was assessed using receiver operating characteristic curve analysis. Among all patients, 39 (58.2%) had poor neurological outcomes. Significant differences in PcsfCO_2_ levels between the groups were observed, with lower PcsfCO_2_ levels on Day 1 showing the highest predictive value at a cutoff of 30 mmHg (area under the curve, sensitivity, and specificity were 0.823, 77.8%, and 79.0%, respectively). These results suggest that PcsfCO_2_ might serve not only as a unique marker for the severity of hypoxic–ischemic brain injury (HIBI), independent of extracorporeal CO_2_ levels, but also as an objective indicator of changes in CSF dynamics. This study highlights the potential prognostic and diagnostic utility of PcsfCO_2_ during TTM in CA survivors, emphasizing its importance in evaluating CSF dynamics and neurological recovery post CA. However, larger multicenter studies are warranted to address potential limitations associated with sample size and outcome assessment methods.

## 1. Introduction

The efficient removal of metabolic waste from the brain is crucial for its health, facilitated by the regulated circulation of cerebrospinal fluid (CSF). CSF, meticulously balanced in composition, maintains brain homeostasis by controlling ion and metabolite levels [1,2,3,4].

Changes in the composition of CSF, including the accumulation of toxic peptides and organic metabolites, occur with aging and conditions such as Alzheimer’s disease and other neurological disorders [1,5,6,7,8,9,10]. This accumulation hampers CSF flow, leading to stasis, which disrupts the delicate balance of fluid dynamics and contributes to CSF stasis. Consequently, these issues affect the gradient of catabolites diffusing from the interstitial fluid to the ventricular CSF, reducing the removal of harmful substances from the brain [1,10,11,12]. Additionally, changes in the expression of aquaporin-4, involved in brain water transport, observed in diseases like Alzheimer’s, also disrupt CSF flow, exacerbating CSF stasis and impacting CSF dynamics [13,14]. Ultimately, these changes affect neuronal function. Therefore, alterations in CSF circulation due to aging or neurological disorders can have adverse effects on neuronal function, potentially impairing cognitive and motor functions [8,10,15].

In some prospective studies involving patients who achieved return of spontaneous circulation (ROSC) after cardiac arrest (CA) and received targeted temperature management (TTM), there was a trend of increased protein concentration in the CSF of patients with poor neurological outcomes compared to those with good neurological outcome [16,17,18,19]. Additionally, an animal study showed changes in the expression of aquaporin, a key channel primarily involved in brain water transport, following exposure to hypoxia [20].

Considering these research findings, we hypothesized that similar to observations in aging and neurological diseases, patients with hypoxic–ischemic brain injury (HIBI) might experience changes in CSF dynamics due to disruption of the blood–brain barrier, leading to the accumulation of toxic peptides, albumin, and organic metabolites in the CSF. Disruptions in CSF flow are expected to particularly interfere with the diffusion of gases, such as carbon dioxide (CO_2_), moving along concentration gradients and through channels like aquaporin [21,22,23]. This interference can be readily assessed through gas analysis. Therefore, the partial pressure of carbon dioxide in the CSF (PcsfCO_2_) might be associated with neurological outcomes and serve as a valuable tool for understanding CSF dynamics in patients with HIBI.

In this study, we aimed to investigate changes in CSF dynamics in the context of HIBI following CA and their impact on neuronal function. Specifically, we evaluated the prognostic value of PcsfCO_2_ for 6-month neurological outcomes in out-of-hospital CA (OHCA) survivors who underwent TTM.

## 2. Materials and Methods

### 2.1. Study Design and Patients

This study was a retrospective analysis of data derived from a prospective observational cohort study conducted at a single tertiary-care center. Data on patients who underwent TTM following OHCA between November 2018 and August 2021 were collected from the registry at Chungnam National University Hospital (CNUH), located in Daejeon, Korea. The study protocol was approved by the Institutional Review Board (IRB) (CNUH IRB2020-04-103) prior to data collection. Written informed consent and approval for the donation of human materials were obtained from the patients’ next of kin. All procedures and protocols adhered to the principles outlined in the Declaration of Helsinki and the International Conference on Harmonization and Good Clinical Practice. The findings are reported following the STROBE criteria. The exclusion criteria were as follows: (1) age < 18 years; (2) ineligibility for TTM (e.g., pre-arrest cognitive impairment, presence of a known terminal illness, intracranial hemorrhage, or active bleeding); (3) ineligibility for lumbar puncture (LP) (e.g., severe cerebral edema, obliteration of basal cisterns, or occult intracranial mass lesion on brain computed tomography); (4) failure to perform LP within 6 h after ROSC; (5) absence of gas analysis at 24 h interval; (6) history of traumatic CA or interrupted TTM; (7) history of extracorporeal cardiopulmonary resuscitation; (8) no next of kin to consent to LP; and (9) refusal of further patient treatment by the next of kin. Out of 100 OHCA survivors with ROSC, 67 were included in this study.

### 2.2. TTM Protocol

Comatose OHCA survivors were managed in accordance with our previously published TTM protocols [24]. TTM was performed using cooling devices (Arctic Sun^®^ Energy Transfer Pads^TM^, Bard Medical, Louisville, CO, USA), with a target temperature of 33 °C maintained for 24 h, and rewarming to 37 °C at a rate of 0.25 °C/h. Core body temperature was monitored using esophageal or bladder temperature probes. Sedatives (midazolam; 0.05 mg/kg intravenous bolus, followed by a titrated intravenous continuous infusion at a rate of 0.05 and 0.2 mg/kg/h) and neuromuscular blocking agents (rocuronium; 0.6 mg/kg intravenous bolus, followed by an infusion at a rate of up to 0.5 mg/kg/h) were used during TTM for sedation and control of shivering, respectively. Anesthetic depth was monitored using ADMS™ (Anaesthetic Depth Monitor for Sedation, Unimedics Co., Ltd., Seoul, Republic of Korea). Fluid resuscitation or vasopressors were administered when necessary to maintain the mean arterial pressure between 65 and 100 mmHg. All other aspects of patient management involved standard intensive care in accordance with our institutional intensive care unit protocol.

### 2.3. CSF and Arterial Blood Gas Analyses

CSF sampling was conducted via lumbar catheter drainage. Lumbar catheter placement was performed using a Hermetic^TM^ lumbar catheter accessory kit (Integra Neurosciences, Plainsboro, NJ, USA) with the patient lying in the lateral decubitus position with the hips and knees flexed. Arterial blood sampling was conducted via radial artery catheterization. Both CSF and arterial blood samples were obtained within 6 h after return of spontaneous circulation (ROSC) (Day 0) and at 24 (Day 1), 48 (Day 2), and 72 h (Day 3) after ROSC. Both sample types were analyzed using the same device (GEM^®^ Premier^TM^ 5000 blood gas analyzer; Werfen, UK) in the Emergency Room of Chungnam National University Hospital. The analysis yielded the following parameters: pH, Arterial oxygen partial pressure (PaO_2_), PaCO_2_, Arterial bicarbonate levels, CSF oxygen partial pressure (PcsfO_2_), PcsfCO_2_, and CSF bicarbonate levels.

### 2.4. Data Collection

The following data were collected from the hospital records: age, sex, first monitored rhythm, etiology of CA, bystander cardiopulmonary resuscitation (CPR), time from CPR to ROSC (low-flow time), time from ROSC to LP (LP time), and CSF sample obtained through LP performed within 6 h after ROSC. These were collected concurrently with arterial blood gas analysis results.

### 2.5. Outcomes

Neurological outcomes were assessed at 6 months after ROSC using the Glasgow–Pittsburgh Cerebral Performance Categories (CPCs) scale, through either face-to-face interviews or structured telephone interviews. Telephone interviews were conducted by an emergency physician who was fully informed about the protocol. Based on the CPC scale scores, outcomes were classified as follows: CPC 1 (good performance), CPC 2 (moderate disability), CPC 3 (severe disability), CPC 4 (vegetative state), or CPC 5 (brain death or death). Good outcome was defined as CPC 1–2 and poor outcome as CPC 3–5. Patients were divided into two groups accordingly. Poor neurological outcome at 6 months post ROSC was the primary outcome of this study.

### 2.6. Statistical Analysis

Categorical variables are summarized as frequencies with percentages, and they were compared between the groups using the chi-square test, with a continuity correction in Fisher’s exact test, as appropriate. Continuous variables are summarized as median values with interquartile ranges, and they were compared between the groups using Mann–Whitney’s U-test. For each time point, receiver operating characteristic curves were plotted and corresponding areas under the curve (AUCs) were determined to evaluate the predictive performance of CSF and arterial blood gas levels for poor neurological outcome, with comparisons among Days 0, 1, 2, and 3. Predictive performance was evaluated using AUC, specificity, and sensitivity with a 95% confidence interval (CI). The optimal cutoff value for predicting poor neurological outcomes 6 months post OHCA was determined using the Youden index. Statistical analyses were performed using IBM SPSS Statistics version 24 (IBM Corp., Armonk, NY, USA) and the MedCalc program, version 15.2.2 (MedCalc Software, Mariakerke, Belgum). *p*-values less than 0.05 were considered to indicate statistical significance.

## 3. Results

### 3.1. Patient Characteristics

Of the 100 OHCA survivors in whom ROSC was achieved, 67 were enrolled in this study. At 6 months after ROSC, 39 (58.2%) patients had poor neurological outcomes (Figure 1). Patient characteristics are shown in Table 1. Compared with those in the poor-neurological-outcome group, patients with good neurological outcomes had a higher incidence of witnessed arrest, were more likely to have a shockable rhythm and cardiac etiology, and had shorter no- and low-flow times. No significant differences between the two groups were noted in any other parameters.

### 3.2. Comparison of CSF and Arterial Blood Gas Analyses between Groups

In the CSF gas analysis, the PcsfCO_2_ on Days 0–3 were significantly lower in the poor- than in the good-neurological-outcome group (Day 0: 27.0 (IQR, 24.0–33.0) vs. 33.2 (IQR, 26.8–38.0), *p* < 0.006; Day 1: 26.5 (IQR, 21.2–30.0) vs. 34.5 (IQR, 30.0–37.2), *p* < 0.001; Day 2: 28.0 (IQR, 19.0–33.0) vs. 35.0 (IQR, 31.9–39.2), *p* < 0.001; Day 3: 27.0 (IQR, 18.5–34.0) vs. 35.0 (IQR, 32.0–39.3), *p* < 0.001) (Table 2). In the arterial blood gas analysis, arterial pH on Day 0 and bicarbonate levels on Days 1 and 2 were significantly lower in the poor- than in the good-neurological-outcome group (Table 3).

### 3.3. Prognostic Performance of PcsfCO_2_

The AUC value of PcsfCO_2_ on Day 1 was highest among those for Days 0–3 (Day 0: AUC 0.755, 95% confidence interval [CI] 0.616–0.863; Day 1: AUC 0.823; 95% confidence interval [CI] 0.692–0.912; Day 2 AUC 0.804, 95% confidence interval [CI] 0.671–0.901; Day 3: AUC 0.760, 95% confidence interval [CI] 0.622–0.806) (Figure 2), with a sensitivity of 77.8% and specificity of 79.0% for predicting poor neurological outcomes. The cutoff value was 30 mmHg (Table 4).

## 4. Discussion

This study represents the first attempt to investigate indicators of changes in CSF dynamics after CA, focusing on gas diffusion, particularly PcsfCO_2_, in relation to the severity of HIBI. Our results revealed a significant association between PcsfCO_2_ levels and neurological outcomes at 6 months after CA.

Among patients with poor neurological outcomes (CPC 3–5), indicating severe HIBI, PcsfCO_2_ levels were markedly lower than those in patients with good neurological outcomes (CPC 1–2), irrespective of arterial PCO_2_ levels. This finding suggests that PcsfCO_2_ may serve as a distinct indicator of HIBI severity, independent of systemic CO_2_ levels.

The discharge of CO_2_ from the brain parenchyma into the CSF likely occurs through gradient-based diffusion [23] and recent studies have revealed specific transport mechanisms such as aquaporin [21]. The permeability of CO_2_ is partially restricted by neurons that demarcate the interstitial fluid from the CSF. Additionally, ependymal cells and perivascular astrocytic endfeet within the brain parenchyma form a relatively less restrictive barrier compared to the tight junctions formed by microvascular endothelial cells [21,22]. CO_2_ originating from the CNS is expelled into the CSF, possibly through paravascular CSF flux or ependymal gap junctions or transport [21,23,25]. Processes such as CO_2_ gas diffusion within the CSF play a crucial role in understanding CSF dynamics. According to Akaishi et al. [26], PcsfCO_2_ levels decrease with age independent of venous blood PCO_2_ levels. This trend can be interpreted as a result of changes in CSF dynamics associated with aging. In this context, our study results suggest that the decrease in PcsfCO_2_ levels observed in patients with HIBI may be a potential surrogate marker reflecting broader alterations in CSF dynamics, in particular, CSF stasis. Considering the potential interference of PcsfCO_2_ with disrupted CSF flow and gas diffusion, it may emerge as a significant indicator of the complex pathophysiological processes occurring post CA.

In numerous studies on aging and neurodegenerative diseases, alterations in CSF, characterized by an increase in protein concentration and compromised clearance mechanisms, have been identified as pivotal factors contributing to disrupted CSF flow [1,5,6,7,8,9,10,11,12]. Additionally, changes in the expression of aquaporin-4, involved in brain water transport, observed in diseases like Alzheimer’s, also disrupt CSF flow, impacting CSF dynamics [13]. These changes can exert profound effects on neuronal performance and may manifest as CSF stasis. The observed disruptions in CSF dynamics, particularly inadequate clearance and potential stasis, have prompted research into the underlying mechanisms. Several studies [16,18,24] have demonstrated that CSF protein concentration increases in patients experiencing HIBI after CA. In a study involving young and aged animals subjected to hypoxia, analysis was conducted on the changes in aquaporin expression, which plays a significant role in CSF production in the brain. This research demonstrated that hypoxia influences aquaporin expression, and it was observed that there was a decrease in the CSF outflow rate and ventricular compliance [20]. Therefore, we hypothesized that the changes in CSF dynamics observed in aging and neurodegenerative diseases, which hinder the efficient clearance of metabolic waste and toxins, may resemble the CSF dynamics in patients with HIBI following CA.

Given the overall impact of disrupted CSF dynamics in HIBI, PcsfCO_2_ monitoring can play a crucial role in clinical settings. Our findings suggest that PcsfCO_2_ may not only serve as a prognostic indicator but also as an early marker of the severity of CSF abnormalities. Integrating PcsfCO_2_ monitoring into routine assessments would enable clinicians to promptly identify patients at risk for neurological deterioration and take immediate measures to alleviate the consequences of impaired CSF circulation. In addition, the comprehensive understanding of the roles of CSF dynamics and PcsfCO_2_ prompts further exploration into targeted therapeutic interventions. Specifically, strategies for optimizing CSF flow and alleviating stasis can be investigated by focusing on regulating PcsfCO_2_ levels. This creates new possibilities for neuroprotective interventions in patients with HIBI post CA.

The strength of our study is the comprehensive evaluation of CSF and arterial blood gas parameters, highlighting differences in PcsfCO_2_ and PaCO_2_ under controlled conditions in patients with HIBI following CA. However, several limitations should be acknowledged. First, this study was conducted at a single center with a small sample size, which may limit the generalizability of our results. Notably, in a previous study, the AUC of CSF metabolites, such as lactate levels, measured 24 h after ROSC for predicting poor neurological outcomes was 0.89, and 30 patients were required to achieve a power of 0.99 at a significance level of 0.05 [27]. Nonetheless, larger multicenter studies are needed to validate our findings. Second, the use of the CPC scale to assess neurological outcomes at 6 months may have introduced subjectivity to the classification. Future studies should consider additional neuroimaging or functional outcome measurements to further enhance the accuracy of prognostic predictions regarding the relationship between CSF stasis and poor neurological outcomes.

## 5. Conclusions

Our study suggests that similar changes observed in neurological disorders such as aging and Alzheimer’s disease may occur in patients with HIBI following CA. This study focused on the increase in CSF protein concentration observed in HIBI patients and changes in aquaporin expression in animals exposed to hypoxia, assuming similar alterations in CSF dynamics. By investigating the PcsfCO_2_, which can be rapidly and easily analyzed through gas analysis, this study aimed to observe the relationship between low PcsfCO_2_ levels and poor neurological outcomes, suggesting that monitoring PcsfCO_2_ could provide early indications of abnormalities in CSF dynamics and contribute to understanding CSF dynamics in HIBI patients. Although our study provides evidence regarding changes in CSF dynamics post-cardiac arrest, larger multicenter studies are needed to improve generalizability and prognostic accuracy.

## Figures and Tables

**Figure 1 brainsci-14-00297-f001:**
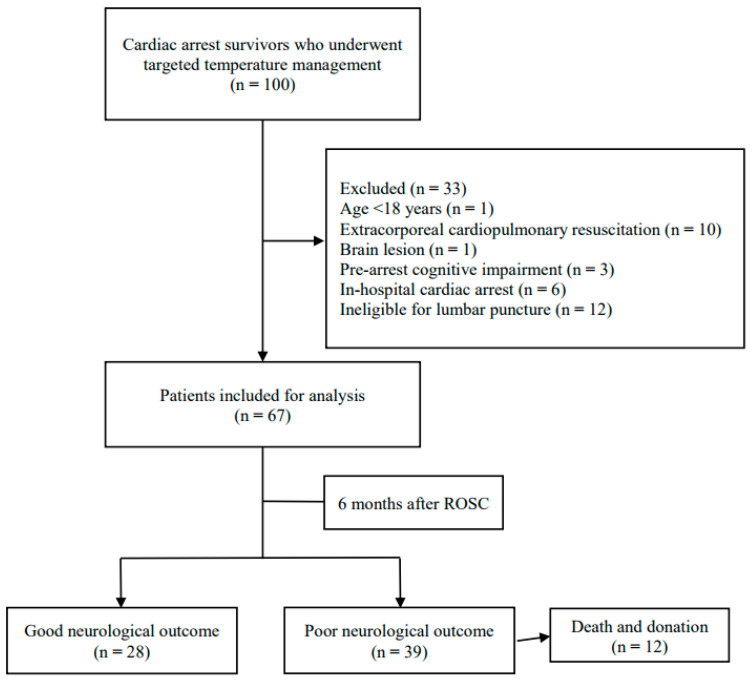
Study flowchart. ROSC, return of spontaneous circulation.

**Figure 2 brainsci-14-00297-f002:**
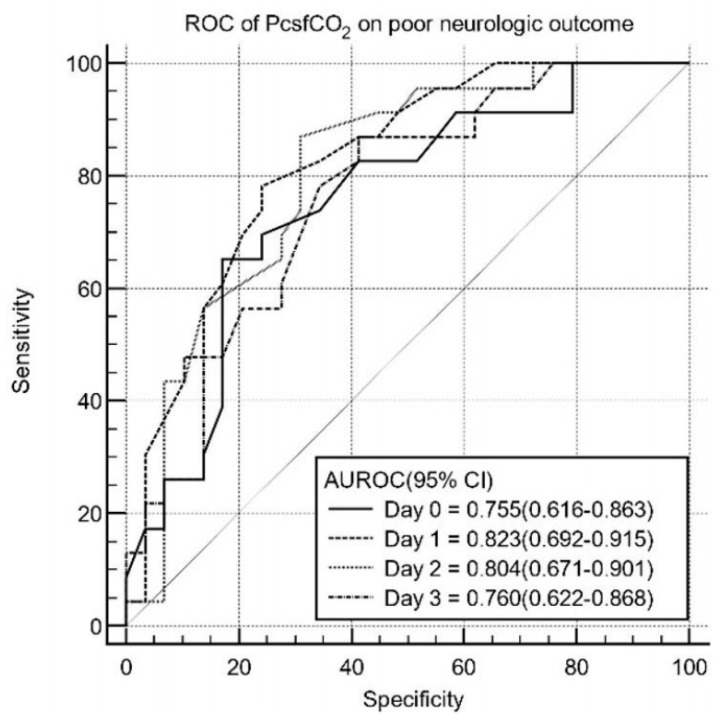
ROC curves of PcsfCO_2_ for predicting poor neurological outcomes 6 months after ROSC. AUC, area under the curve; PcsfCO_2_, cerebrospinal fluid partial pressure of carbon dioxide; ROC, receiver operating characteristic; ROSC, return of spontaneous circulation; Day 0, immediately after ROSC; Day 1, 24 h after ROSC; Day 2, 48 h after ROSC; Day 3, 72 h after ROSC.

**Table 1 brainsci-14-00297-t001:** Baseline demographic and clinical characteristics.

Characteristics	Cohort (N = 67)	Good Outcome (*n* = 28)	Poor Outcome (*n* = 39)	*p*-Value
Demographic characteristics				
Age, years (IQR)	60.0 (41.0–70.0)	58.5 (43.5–69.6)	56.8 (38.0–72.0)	0.652
Sex, male, *n* (%)	49 (73.1)	23 (82.1)	26 (66.7)	0.177
Cardiac arrest characteristics				
Witnessed arrest, *n* (%)	38 (56.7)	23 (82.1)	15 (39.5)	<0.001 *
Bystander CPR, *n* (%)	47 (70.1)	23 (82.1)	24 (61.5)	0.104
Shockable rhythm, *n* (%)	19 (28.4)	16 (57.1)	3 (7.7)	<0.001 *
Cardiac etiology, *n* (%)	27 (40.3)	18 (64.3)	9 (23.1)	<0.001 *
No flow time, min, median (IQR)	1.0 (0.0–13.5)	1.0 (0.0–1.3)	9.5 (0.0–25.5)	0.002 *
Low flow time, min, median (IQR)	20.0 (9.0–28.3)	10.0 (8.0–19.0)	26.0 (19.0–42.0)	<0.001 *
ROSC to LP time, h(IQR)	4.5(3.2–6.0)	4.1(3.0–5.9)	4.7(4.0–6.0)	0.150

CPR, cardiopulmonary resuscitation; IQR, interquartile range; LP, lumbar puncture; ROSC, return of spontaneous circulation. * *p* < 0.05 was considered statistically significant.

**Table 2 brainsci-14-00297-t002:** Association of CSF pH, PcsfO_2_, PcsfCO_2_, and HCO_3_ with poor neurological outcome.

Origin of Sample	Cohort (N = 67)	Good Outcome (*n* = 28)	Poor Outcome (*n* = 39)	*p*-Value
CSF pH median (IQR)				
Day 0	7.43 (7.39–7.49)	7.45 (7.40–7.52)	7.41 (7.385–7.48)	0.207
Day 1	7.47 (7.41–7.69)	7.49 (7.45–7.92)	7.46 (7.39–7.64)	0.251
Day 2	7.45 (7.38–7.91)	7.62 (7.42–8.05)	7.43 (7.37–7.68)	0.130
Day 3	7.53 (7.41–7.88)	7.60 (7.42–7.80)	7.47 (7.41–7.82)	0.654
CSF PcsfO_2_ median (IQR)				
Day 0	156.0 (116.0–190.0)	153.5 (127.8–165.0)	148.0 (131.0–158.0)	0.703
Day 1	138.0 (120.0–159.0)	151.0 (121.0–170.1)	149.0 (122.5–170.3)	0.940
Day 2	123.5 (106.5–136.3)	132.0 (122.5–165.0)	141.0 (125.0–163.0)	0.713
Day 3	125.0 (104.5–143.5)	143.0 (122.4–170.0)	151.0 (127.5–166.5)	0.699
CSF PcsfCO_2_ median (IQR)				
Day 0	37.4 (31.0–43.5)	33.2 (26.8–38.0)	27.0 (24.0–33.0)	0.006 *
Day 1	38.0 (35.0–43.5)	34.5 (30.0–37.2)	26.5 (21.2–30.0)	<0.001 *
Day 2	42.0 (36.8–45.3)	35.0 (31.9–39.2)	28.0 (19.0–33.0)	<0.001 *
Day 3	42.0 (38.1–45.6)	35.0 (32.0–39.3)	27.0 (18.5–34.0)	<0.001 *
CSF HCO_3_ median (IQR)				
Day 0	20.4 (18.5–23.3)	22.2 (20.0–23.7)	19.4 (15.5–20.9)	0.003 *
Day 1	22.0 (19.9–25.0)	23.8 (21.4–27.4)	21.2 (18.6–23.2)	0.019
Day 2	22.2 (20.2–34.3)	24.3 (21.8–27.6)	20.6 (18.2–22.5)	0.001 *
Day 3	23.5 (21.5–26.3)	24.6 (22.7–36.4)	22.3 (18.5–26.4)	0.117

CSF, cerebrospinal fluid; HCO_3_, bicarbonate; IQR, interquartile range; PcsfO_2_, cerebrospinal fluid partial pressure of oxygen; PcsfCO_2_, cerebrospinal fluid partial pressure of carbon dioxide; Day 0, immediately after ROSC; Day 1, 24 h after ROSC; Day 2, 48 h after ROSC; Day 3, 72 h after ROSC; ROSC, return of spontaneous circulation. * *p* < 0.05 was considered statistically significant.

**Table 3 brainsci-14-00297-t003:** Association of arterial pH, PaO_2_, PaCO_2_, and HCO_3_ with poor neurological outcome.

Origin of Sample	Cohort (N = 67)	Good Outcome (*n* = 28)	Poor Outcome (*n* = 39)	*p*-Value
Arterial pH median (IQR)				
Day 0	7.30 (7.25–7.40)	7.33 (7.29–7.40)	7.27 (7.20–7.37)	0.067
Day 1	7.35 (7.28–7.40)	7.37 (7.30–7.41)	7.32 (7.27–7.40)	0.173
Day 2	7.39 (7.35–7.42)	7.41 (7.38–7.43)	7.38 (7.33–7.40)	0.013 *
Day 3	7.40 (7.35–7.44)	7.41 (7.34–7.45)	7.39 (7.35–7.44)	0.535
Arterial PO_2_ median (IQR)				
Day 0	156.0 (116.0–190.0)	151.5 (123.8–193.0)	157.0 (104.0–189.0)	0.844
Day 1	138.0 (120.0–159.0)	139.0 (114.8–153.5)	138.0 (122.0–159.0)	0.760
Day 2	123.5 (106.5–136.3)	127.0 (113.5–138.3)	115.0 (102.0–134.8)	0.159
Day 3	125.0 (104.5–143.5)	123.0 (106.0–154.0)	125.5 (104.0–135.5)	0.766
Arterial PCO_2_ median (IQR)				
Day 0	37.4 (31.0–43.5)	36.5 (31.3–40.0)	38.0 (31.0–48.0)	0.277
Day 1	38.0 (35.0–43.5)	38.9 (35.0–44.0)	38.0 (35.0–42.0)	0.670
Day 2	42.0 (36.8–45.3)	42.3 (36.0–46.1)	42.0 (37.8–45.3)	0.907
Day 3	42.0 (36.8–45.3)	40.0 (38.0–45.3)	42.0 (37.8–45.9)	0.856
Arterial HCO_3_ median (IQR)				
Day 0	19.8 (15.7–22.0)	21.1 (17.6–22.8)	18.6 (15.0–20.8)	0.095
Day 1	22.7 (19.6–24.4)	23.7 (21.2–24.7)	21.7 (17.9–24.4)	0.028 *
Day 2	25.3 (22.8–26.7)	25.7 (23.9–27.3)	24.5 (22.0–26.0)	0.037 *
Day 3	25.6 (23.9–27.7)	26.3 (24.5–28.3)	24.6 (23.6–26.6)	0.055

HCO_3_, bicarbonate; IQR, interquartile range; PO_2_, partial pressure of oxygen; PCO_2_, partial pressure of carbon dioxide; Day 0, immediately after ROSC; Day 1, 24 h after ROSC; Day 2, 48 h after ROSC; Day 3, 72 h after ROSC; ROSC, return of spontaneous circulation. * *p* < 0.05 was considered statistically significant.

**Table 4 brainsci-14-00297-t004:** Association of PcsfCO_2_ with poor neurological outcome.

Sample	Time	AUC (95% CI)	*p* Value	Cutoff	Sensitivity (95% CI)	Specificity (95% CI)	PPV/NPV
PcsfCO_2_	Day 0	0.755 (0.616–0.863)	0.002 *	31 (mg/dL)	60.7 (40.6–78.5)	74.4 (57.9–87.0)	68.0/57.8
	Day 1	0.823 (0.692–0.915)	<0.001 *	30 (mg/dL)	76.9 (56.4–91.0)	78.9 (62.7–90.4)	82.2/71.3
	Day 2	0.804 (0.671–0.901)	<0.001 *	30 (mg/dL)	88.0 (68.8–97.5)	67.7 (47.8–80.9)	91.0/80.4
	Day 3	0.760 (0.622–0.806)	<0.001 *	29 (mg/dL)	87.0 (66.4–97.2)	58.6 (38.9–76.5)	90.2/76.7

AUC, area under the curve; CI, confidence interval;PcsfCO_2_, cerebrospinal fluid partial pressure of carbon dioxide; Day 0, immediately after ROSC; Day 1, 24 h after ROSC; Day 2, 48 h after ROSC; Day 3, 72 h after ROSC; ROSC, return of spontaneous circulation; PPV, positive predictive value; NPV, negative predictive value. * *p* < 0.05 was considered statistically significant.

## Data Availability

The data presented in this study are available on request from the corresponding author. The data are not publicly available, because of ethical concerns.

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
