# Peer review of "Cerebrospinal Fluid Dynamics and Partial Pressure of Carbon Dioxide as Prognostic Indicators in Hypoxic–Ischemic Brain Injury Following Cardiac Arrest"

_brainsci, 2024, doi:10.3390/brainsci14030297_

Round 1

Reviewer 1 Report

Comments and Suggestions for Authors

The study investigates the prognostic value of cerebrospinal fluid (CSF) dynamics, specifically the partial pressure of carbon dioxide (PcsfCO2), in hypoxic-ischemic brain injury (HIBI) following cardiac arrest (CA). The authors retrospectively analyzed 67 CA survivors who underwent targeted temperature management and examined their 6-month neurological outcomes and CSF and arterial blood gas parameters. The study found significant differences in PcsfCO2 levels between patients with good and poor neurological outcomes, with lower PcsfCO2 levels on Day 1 having the highest predictive value for poor outcomes. The findings suggest that PcsfCO2 could serve as a distinct marker for the severity of HIBI, independent of systemic CO2 levels. Monitoring PcsfCO2 levels may have prognostic value and provide early indications of CSF abnormalities in CA survivors.

The similarity ratio is very high (37%), and is to be reduced. 

. The abstract lacks a clear statement of the research objectives and the significance of the study. It would be helpful to provide a concise statement of the research aim and the implications of the findings for clinical practice.

. The methodology section needs further elaboration. It should include information on the study design, inclusion and exclusion criteria, data collection methods, and statistical analyses performed.

. The results section does not provide specific numerical data or statistical measures, such as p-values or confidence intervals. Including these details would enhance the clarity and reliability of the findings.

. The introduction provides a general overview of CSF dynamics but lacks a clear rationale for studying PcsfCO2 as a prognostic indicator. It would be beneficial to provide more background information on the relationship between CSF dynamics, PcsfCO2, and neurological outcomes in HIBI patients.

. The keywords could be more comprehensive and representative of the study's focus. Consider adding terms such as "cerebral hypoxia," "neurological recovery," or "out-of-hospital cardiac arrest."

. The conclusion section should summarize the key findings and their implications clearly. Additionally, it would be valuable to discuss any limitations of the study and suggest future research directions.

Author Response

Your review has been immensely helpful for my research and paper. I appreciate your thorough examination.

  1. As you suggested, I have revised the abstract to clearly state the research objectives and the significance of the findings.
  2. 2. I have provided clearer exclusion criteria in the methodology section, and elaborated more on the data collection methods and statistical analysis.
  3. 3. In the results section, I have added specific numerical data and statistical measures to enhance clarity and reliability.
  4. 4. I have added explanations in the Introduction regarding the rationale for studying PcsfCO2 and provided additional background information on changes in cerebrospinal fluid (CSF) dynamics in patients with hypoxic-ischemic brain injury (HIBI).  
  5. I have included "cerebral hypoxia," "neurological recovery," and "OHCA" in the keywords to better represent the focus of the study. 
  6. In the conclusion, I have summarized the key findings and proposed future research directions.

Reviewer 2 Report

Comments and Suggestions for Authors

Jeon et al investigated the association between CSF partial pressure of carbon dioxide (PcsfCO2) and neurological outcomes in 100 cardiac arrest (CA) survivors. This study suggests that PcsfCO2 could serve as a potential biomarker for assessing severity of hypoxic-ischemic brain injury (HIBI). While the findings hold significance in HIBI management and are potentially interesting for the readers of this journal, additional work is needed to substantiate their findings.

Major points:

1. In section 3.3, please incorporate a classification model that utilizes the data of PcsfCO2 from 0-3 days collectively. Present the Receiver Operating Characteristic Area Under the Curve (ROC-AUC) for this combined model.

2. Please compare the PcsfCO2 index with other CSF indices in relation to neurological outcomes. Provide an explanation for why PcsfCO2 is regarded as a superior index compared to other CSF indices.

3. Please discuss the reasons supporting the potential of PcsfCO2 as a biomarker for indicating cerebrospinal fluid (CSF) disruption.

4. In conclusion, the author asserts the independence of the PcsfCO2 index from arterial CO2 levels. Please perform an analysis to substantiate this claim.

Minor points:

5. On line 25, maintain consistency in the format of the metrics.

6. Please Incorporate the participant description from section 3.1 into section 2.2.

7. On line 104, please clarify if PCO2 is same as PcsfCO2.

8. In Table 1, provide clarification for the absence of the age information in the poor outcome group.

9. In Table 2, if statistical significance is attributed to P<0.05, please explain why the CSF HCO3 on day 1 is not regarded as significant.

Author Response

Your review has been invaluable for my research and paper. I appreciate your thorough examination.

  1. In Section 3.3, we supplemented the analysis by providing the ROC-AUC for PcsfCO2 from 0-3 days.
  2. Unlike other biomarkers that may require days or complex analyses, PcsfCO2 can be immediately assessed using rapid and straightforward gas analysis, potentially increasing its utility.
  3. While CO2 can diffuse based on gradient differences, recent research has identified Aquaporin as a transporter for CO2. Animal studies have shown that exposure to hypoxia leads to changes in aquaporin expression, suggesting that PcsfCO2 could indicate changes in CSF dynamics in patients with HIBI after cardiac arrest. We addressed this in the discussion section.
  4. We ensured consistency in the format of the metrics on line 25.
  5. In Section 2.2, we included a description of the participants from Section 3.1.
  6. We clearly distinguished between PaCO2 (PCO2 in arterial blood) and PcsfCO2 (PCO2 in CSF).
  7. Double-checked the missing age information for patients with poor neurological outcomes in Table 1.

Reviewer 3 Report

Comments and Suggestions for Authors

In this study, researchers aimed to evaluate the relationship of partial CO2 pressure in cerebrospinal fluid with the prognosis of hypoxic-ischemic brain injury after cardiac arrest. I think the subject of the research is original. The method of the research is properly designed. The results are clearly reported. The discussion has been made appropriately in the light of the literature. The result of the research can be applied and used in clinical practice. On the other hand, I have a question and a correction suggestion for this manuscript.

1)      Were there any differences between CPC1 and CPC2 in the good outcome group?

2)      The mean age of the poor outcome group is not written in Table 1. I think it was overlooked

Author Response

Your review has been immensely helpful for my research and paper. I appreciate your thorough examination.

  1. We found no significant difference in PcsfCO2 between CPC 1 and CPC 2 within the good neurological outcome group (p < 0.001). However, due to the small sample size (n = 28), this may not be a conclusive result, and there could be errors. It would be necessary to reevaluate this when conducting larger studies in the future.
  2. I realized there was an oversight regarding the missing average age for patients with poor neurological outcomes. I have double-checked this information.

Round 2

Reviewer 2 Report

Comments and Suggestions for Authors

Following a comprehensive review of the revisions made by the authors, I have only one minor suggestion about citation on the correlation between changes in the composition of CSF and Alzheimer’s disease on line 70. I recommend including the latest literature by Lin et al. in Annals of Neurology in 2021 (https://doi.org/10.1002/ana.26134), as it presents the most recent findings in this area. This additional citation would bolster the evidence supporting the association between CSF composition and Alzheimer's disease.

Author Response

Thanks to your advice, I was able to bolster the evidence supporting the association between CSF composition and Alzheimer's disease. I am truly grateful. I have revised my manuscript by attaching references as per your advice.